# Multi-Scale Influencing Factors and Prediction Analysis: Dongxing Port–City Relationship

**DOI:** 10.3390/ijerph19159068

**Published:** 2022-07-25

**Authors:** Bin Wu, Guanhai Gu, Wenzhu Zhang, Liguo Zhang, Rucheng Lu, Caiping Pang, Jixian Huang, Hongxin Li

**Affiliations:** School of Natural Resources and Surveying and Mapping, Nanning Normal University, Nanning 530011, China; wubin@nnnu.edu.cn (B.W.); lurucheng@163.com (R.L.); p18333773461@163.com (C.P.); x18877174710@163.com (J.H.); lihongxin0115@163.com (H.L.)

**Keywords:** port–city relationship, improved dynamic concentration index, multi-scale factors, dynamic simulation, port–city development, Dongxing City

## Abstract

This study calculates the dynamic concentration index, explores the evolution of the relationship between the Dongxing port and city, and predicts its future. The results indicate that the relationship between the port and city has three development stages, namely the low-level balanced development stage (2001–2008), the port development stage (2009–2014), and the urban development stage (2015–2019). Based on the country (China and Vietnam), province (Guangxi Zhuang Autonomous Region), district (Fangchenggang City), county (Dongxing City), and individual (resident) levels, a multi-scale index system of influencing factors was developed. The impulse response function model analyzed the influential factors in the relationship between port and city development. The influence is as follows: country (China) > country (Vietnam) > county > individual > city > province. Finally, the relationship between port and city development was predicted using an auto-regression differential moving average model. It is expected that Dongxing City will gradually transition from a port- and city-dominated stage to a new stage of coordinated development. Thus, by improving the proportion of the secondary and tertiary industries, managing the population density, introducing foreign capital, enhancing the innovation level, and improving the traffic facilities, high-quality development in Dongxing port–city can be achieved.

## 1. Introduction

China has a land border of 22,000 km and a coastline of 18,000 km. It borders fourteen countries on land and faces eight countries by sea. As of 31 December 2020, the state council had approved the opening of a total of 313 ports to the outside world, including 82 highway ports, 129 water ports, 22 railway ports, and 80 airports. Land border areas have 140 ports open to the outside world, of which 70 are highway ports, 31 are water ports, 13 are railway ports, and 26 are airports. With the deepening of the national strategic concept of the “Belt and Road Initiative” to promote endorsement of the Regional Comprehensive Economic Partnership agreements (RCEP), domestic and international strategies concerning frontier ports and cities have also shifted. Economic resources at the regional level, global configuration, and implementation of more effective linkages of regional and global economic growth are also developing.

Regarding the nodes and hubs of communication with the outside world, border ports are far from the national center, vulnerable to geographical factors, and possess the features of international passenger and freight transport. A port–city system has three characteristics: (1) It is external-facing; therefore, it connects domestic and foreign economic exchanges and represents the intersection of international trade; (2) it is comprehensive, and as a multi-level, multi-scale, and multi-regional complex, the port economy affects the regional, city, and national economies, and (3) it is interdependent as the economic activities of the port depend on the development of the port itself and the economic activities of the city with the outside world. This study used Dongxing port–city as the research phenomenon and analyzed the development and evolutionary relationships between the port and the city. The study explored its evolutionary mechanism and suggested new ideas to promote the sustained and steady development of the China–Vietnam border trade.

Much domestic and foreign literature has studied the port–city relationship. The main research topics are as follows. First, regarding the interaction between ports and cities, the main structure and development dynamics of port cities have been studied on a global scale [1,2]. Previous studies have analyzed the scale of international trade, structural changes in port activities, and the influence of port industrialization, traffic origin, and destination factors on the relationship between port cities [3,4]. In addition, the general economic growth model of ports and cities has been summarized from the dynamic structure and economic characteristics of ports and the strength of the relationship between ports and cities [5,6,7,8,9]. The port–city spatial system evolution model was adopted to discuss the change rule of the intensity curve of the port–city relationship [10]. 

Second, based on the quantitative analysis of the relationship between ports and cities, the dynamic concentration index (DCI) has been used to classify these relationship types. The relative concentration index (RCI) has been used to measure the organizational relationship between the port areas and human settlements [11,12]. These indices reveal the evolution characteristics of the port–city relationship in modern China [7], indicate the quantitative analysis of the functions of the port–city, and demonstrate the general law of the development of the port–city functional relationship [13]. 

Third, regarding the evolution mechanism of the relationship between ports and cities, the development of the relationship between ports and cities has been explored from multiple dimensions [14]. The RCI was used to analyze the unbalanced development of the relationship between ports and cities in northeast China [15]. Through the six-stage model of port–city development, the spatial evolution process from symbiosis to separation, and thereafter demonstrated port–city redevelopment [16]. 

Fourth, regarding the coordination between port–city size and function used the average tonnage of cargo per resident to measure the importance of the transportation function and city size of American port cities [17]. The relationship matrix of port–city functions was proposed to explain the spatial and functional characteristics of European port cities [18]. Based on the evolutionary trajectory of the port–city coordinated development mode, the comprehensive development index was proposed [19], and the life cycle of port–city relationship development was studied using systems theory [20,21]. 

Fifth, regarding the relationship between the port and city dynamic simulation, based on the system dynamic causality diagram, a series of markers of the harbor city development factors was selected [19]. To evaluate the relationship between urban development status and various indicators, a logistic model was used to simulate urban expansion by considering spatio-temporal dimensions [22,23]. The cellular automata model was used to dynamically simulate the urban geospatial pattern and evolution process [14].

The existing research has mostly analyzed the evolution of the port–city relationship on a single scale, whereas the present study systematically analyzes the mutual feed mechanism and dynamic simulation of the port–city relationship from a multi-scale perspective. First, the improved DCI is used to calculate the cross-index between the ports and cities. The development stage of the Dongxing port–city relationship is divided according to the number of passengers and shipments at the Dongxing port and the population and GDP of Dongxing City. Second, a comprehensive index system is constructed from the perspectives of the country (China and Vietnam), province (Guangxi Zhuang Autonomous Region), city (Fangchenggang City), county (Dongxing City), and individual (residents). The impulse response function based on the VAR model is used to analyze the multi-scale influencing factors of the port–city relationship development. Finally, the autoregressive integrated moving average (ARIMA) model is used to predict the evolution process of the port–city relationship in the future. Some policy suggestions are presented for the coordinated development of the Dongxing port and city.

## 2. Materials and Methods

### 2.1. Research Area and Data Sources

Dongxing, located at the southwestern end of China’s coastline, is the most convenient land and sea channel to Vietnam and the Association of Southeast Asian Nations (ASEAN) and is an important commercial port for trade between China and Southeast Asia. In 1958, it was listed as a national first-class port, and by 1996, it was established as a county-level city with an administrative area of 590 km^2^. In 2000, China and Vietnam issued a joint statement regarding all-around cooperation in the new century, and bilateral economic and trade relations have developed rapidly and steadily. After 20 years of construction, Dongxing port has begun to take shape. In 2020, Dongxing had a GDP of USD 1.16 billion, a registered population of 160,200, and a total foreign trade volume of USD 3.67 billion. With the deepening of the national “Belt and Road Initiative,” development in China is playing an increasingly important role. This development entailed implementing an ASEAN free trade area and authorizing the RCEP, with the Dongxing City development experimental zone as a key national large port cross-border economic cooperation zone. The pattern of opening up to the outside world played an important role in these changes.

This study systematically collected the economic and social development data of Dongxing port and Dongxing City. The primary data are from the Dongxing Statistical Yearbook of National Economic and Social Development, the China County-level Statistical Yearbook, Fangchenggang Statistical Bulletin of National Economic and Social Development, the Guangxi Zhuang Autonomous Region Statistical Yearbook of National Economic and Social Development, Guangxi Zhuang Autonomous Region Statistics Bureau (http://tjj.gxzf.gov.cn/, accessed on 21 March 2022), the National Office of Port Administration website (http://gkb.customs.gov.cn/gkb/2691117/index.html, accessed on 21 March 2022), China’s Belt and Road Portal (https://www.yidaiyilu.gov.cn/, accessed on 21 March 2022), and the United Nations trade statistics database (https://comtrade.un.org, accessed on 21 March 2022).

### 2.2. Research Methods

#### 2.2.1. Modified Dynamic Concentration Index

This study uses the Modified Dynamic Concentration Index (MDCI) to represent the relationship between port and urban development [14]. The DCI value of the urban development scale and port development scale index is calculated to reflect the development relationship between the ports and cities:(1)$$D_{e}=\left[\sqrt[{n-1}]{\frac{A_{n}}{A_{1}}}-1\right]/\left[\sqrt[{n-1}]{\frac{B_{n}}{B_{1}}}-1\right]$$
(2)$$D_{i}=\left[\frac{A_{n}-A_{1}}{\left(n-1\right)\times \sum_{i=1}^{n}A_{i}}\right]/\left[\frac{B_{n}-B_{1}}{\left(n-1\right)\times \sum_{i=1}^{n}B_{i}}\right]$$
(3)$$DCI=\alpha D_{e}+\beta D_{i}$$
(4)$$MDCI=W_{cp}\times DCI_{cp}+W_{cg}\times DCI_{cg}+W_{tp}\times DCI_{tp}+W_{tg}\times DCI_{tg}$$
where *D_e_* represents the elasticity coefficient of port-urban development, which refers to the ratio of the average growth rate of the port development level (freight or passenger transport) to the average growth rate of the urban development level (population or GDP) in a certain period. *D_i_* is the relative concentration index of the port–city development and refers to the proportion of the average growth of the port development level (freight or passenger transport) to the average growth of the urban development level (population or GDP) in a certain period.

#### 2.2.2. Impulse Response Function Based on VAR Model

The impulse response function is mainly used to measure the impact of a standard deviation shock (pulse) of a random disturbance term of an endogenous variable in the model on the current and future values of all endogenous variables of the model:(5)$$Y_{1t}=\alpha_{11}Y_{1t-1}+\alpha_{12}Y_{2t-1}+\varepsilon_{1t}$$
(6)$$Y_{2t}=\alpha_{12}Y_{2t-1}+\alpha_{22}Y_{2t-1}+\varepsilon_{2t}$$
where *Y*_1*t*_ represents the k dimension endogenous variable vector, *t* represents time, and *ε_t_* is the random disturbance term. If an impact on *ε_t_* changes *Y_t_*, then the next value of *Y*_2*t*_ will also change through the model’s action. If affected by hysteresis, *Y*_2*t*_ will affect the change of *Y_t_*’s future value. In this study, the impulse response function is mainly used to discuss the possible impact of a shock on the port–city relationship on subjects of different scales.

#### 2.2.3. PCA (Principal Component Analysis)

By reducing the dimension of the data, PCA transforms a variety of variable data into a few independent principal components, namely comprehensive variables. The cumulative variance contribution rate of the principal component can guarantee most of the characteristic information of the variable data, and its formula is as follows:(7)$$\{\begin{array}{c} K_{1}=c_{11}y_{1}+c_{12}y_{2}+\cdots +c_{1n}y_{n}\\ K_{2}=c_{21}y_{1}+c_{22}y_{2}+\cdots +c_{2n}y_{n}\\ \cdots \\ K_{n}=c_{n1}y_{1}+c_{n2}y_{2}+\cdots +c_{nn}y_{n} \end{array}$$
where $y_{1}$, $y_{2}$, $y_{3}$, …, $y_{n}$ is the initial variable; $K_{n}$ is the main component, and $c_{ij}$ is the principal component coefficient.

#### 2.2.4. Stationary Test Model

The characteristic equation is used to judge the stability of the VAR model. If the model is stable, all the characteristic values will be inside the unit circle (the unit circle is a circle in the coordinate system with the origin as the center of the circle, radius 1, real number axis as the horizontal axis and virtual number axis as the vertical axis), and its principle is as follows:(8)$$(I-\prod_{l}L)\;Y_{t}=\mu +\upsilon_{t}$$
where $\prod_{l}$ is the parameter matrix, μ is the column vector of the constant term, and $\upsilon_{t}$ is the random error vector.

#### 2.2.5. Autoregressive Differential Moving Average Model (ARIMA)

This model consists of three parts: autoregression (AR), difference (I, which stands for integration), and moving average (MA). The time series must be stable before the econometric model can be established. The unit root test is conducted on the time series. If the time series is non-stationary, it needs to be transformed into the stationary series by the difference and then transformed into the stationary series after several differences, which is known as multi-order integration. The ARIMA model is denoted as ARIMA (P, D, Q), where P is the number of autoregression terms, D is the number of differences (order) to convert it to a stationary sequence, and Q is the number of terms of the moving average.

## 3. Results

### 3.1. Development of the Relationship between Ports and Urban Development in Dongxing City

#### 3.1.1. Development Stages of the Relationship between Ports and Cities

From the data regarding passenger quantity, Dongxing port quantity, and the Dongxing City population and GDP, the cross DCI index of each index is calculated according to the above Equations (1)–(3). Four groups of cross indices are obtained: the DCI index of the quantity and population (DCIcp), quantity and GDP (DCIcg), number of arrivals and population (DCItp), and number of arrivals and GDP (DCItg). The weight of the cross DCI index (Table 1) was calculated using the entropy weight TOPSIS method; the MDCI value for each year was obtained (Figure 1), and the curve was drawn.

Based on the analysis of the above DCI index and MDCI index each year, combined with the actual data change of the Dongxing port and the city scale and the improved dynamic concentration index change curve, the developing relationship between Dongxing port and the city can be divided into three stages: 2001–2008, 2009–2014, and 2015–2019 corresponding to n = 8, n = 6, and n = 5, respectively. DCI and MDCI (Table 2) were cross-calculated according to Equations (1)–(4), and comprehensive indices representing the port–city relationship in different development stages were obtained.

#### 3.1.2. Analysis of the Characteristics of Ports and Cities in Different Development Stages

The first stage (2001–2008) was a low-level, balanced development. Both the MDCI index and MDe are positive and greater than one, while the MDi is greater than 0 and slightly greater than 1. This indicates that the growth rate of the port and the growth rate of scale are slightly greater than those of the cities. 

For the ports, the total shipment quantity is increasing, and the growth rate is fast, reaching 93.99% in 2008. The pace and volume of passenger traffic have steadily increased.Regarding the cities, the GDP had an increasing trend, but it fluctuated in the medium term. The total population remained roughly the same.The DCI index demonstrates that the growth rate of the port transit and size is lower than that of the urban population but larger than that of the GDP. The growth rate of the port passenger volume and size is lower than that of the urban population but larger than that of the GDP.

In the second stage (2009–2014), port development was slightly faster than urban development. The MDCI and MDi are also positive and greater than one. This indicates that the ports’ growth rate and scale are greater than those of the cities, leading to the following inferences: (1) in the ports, the quantity of goods is growing at a low speed. The growth rate of the passenger volume changes from negative to positive, and the speed gradually increases; (2) in the urban areas, the population growth rate remains low. The total GDP has been increasing, but the growth rate has slowed down. (3) From the DCI index, the growth rate of the port transit quantity and size is lower than that of the urban population but larger than that of the urban GDP. The growth rate of the port passenger volume and size is lower than that of the urban population and larger than the urban GDP.

In the third stage (2015–2019), urban development is faster than port development. Both the MDCI index and MDe are negative and equal to one; however, MDi is positive but less than one. This indicates that the port growth rate and scale growth rate are lower than that of the cities: (1) in the ports, the growth rate and total quantity of the shipments gradually increased, and the total volume and speed of the passenger traffic increased. (2) Regarding the cities, the population keeps a low growth rate, and the total GDP and speed increase steadily. (3) The DCI index indicates that the growth rate of port transit quantity and size is less than that of the urban population and GDP.

### 3.2. Multi-Scale Analysis of Influencing Factors on the Relationship between Port and City

#### 3.2.1. Overview of Multi-Scale Subjects

There are varying degrees of impact on the development of city relationships. From the perspective of national (China and Vietnam), provincial (Guangxi Zhuang autonomous region), district (Shenzhen), county (Dongxing City), and individual (residents) dimensions, a comprehensive index system was developed (Table 3). This study analyzes the mechanism of different factors in the evolution of the Dongxing port–city relationship from spatial and temporal dimensions.

#### 3.2.2. Model Stability Test

The principal components affecting the development of the port–city relationship in five dimensions were extracted, and the principal component analysis was used for information enrichment study. A set of data representing the principal components at different scales was obtained (Table 4). KMO values from the national scale to the individual scale were all greater than 0.75, indicating that the data can be used for principal component analysis research. Bartlett sphericity test was performed on the data of five scales. From the perspective of total variance interpretation, the values of each scale were all greater than 90%, thereby indicating that the data can better explain the influencing factors of the port–city relationship [24]. Stationarity test on the influencing factors using the Formula (8) for each scale data for system stability analysis as shown in the VAR system stability discriminant graph (Figure 2). The results indicate that the eigenvalues of the five dimensions are within the unit circle, therefore it is confirmed that this model is stable.

#### 3.2.3. Analysis of Impulse Response Function

The influence of the impulse response function curves to scale is obtained from the national, provincial, district, county, and individual levels. These indicators were selected for economic and social development to construct the impulse response function between the large port–city MDCI index (Figure 3) in order to reveal the multiple factors under the different scales of the port–city relationships [25].

The impulse response function describes the influence of each endogenous variable in the VAR model on itself and other endogenous variables. The impulse response function also serves to observe the influence of each variable in the model on the response over time. From the curve fluctuation of each scale, subjects at different scales have different effects on the port–city relationship. The influence of different scales is as follows: country (China) > country (Vietnam) > county > individual > city > province.

According to the impulse response function on the national scale, the overall impact of both China and Vietnam is larger, whereas the impact magnitude of China is larger than that of Vietnam. The impact of the international tourism revenue and expenditure is greatest for Vietnam. From 2001 to 2007, the influence of the international tourism income and expenditure was relatively stable, but the curve fluctuation increased in the later period and even appeared as an extreme value (negative value) in 2009. The other indicators greatly fluctuated from 2001 to 2013 and then gradually stabilized. The overall curve for China fluctuated greatly from 2001 to 2015 and, thereafter, began to stabilize. Import and export indicators fluctuated the most and had a negative impact in 2009, 2015, and 2016. National factors have the most significant influence on the Dongxing port–city relationship, which is closely related to the geographical location, economic growth, industrial structure, and development background of the region. Considering border trade cities, Dongxing port undertakes the land transport task of China–Vietnam trade, which is closely related to macroeconomic factors, such as China’s economic development level, urbanization growth rate, and development and opening of the border areas. Vietnam’s economic development directly impacts the scale of cargo transportation and the number of people entering and exiting the Dongxing port, and this further affects the development and evolution of the relationship between ports and cities.

On the county scale, the impact was large in the early stages and tended to be stable in the later stages. From 2001 to 2005, the GDP and value added of the secondary and tertiary industries in Dongxing City greatly fluctuated and demonstrated a negative impact in 2003 and 2004. The other indicators fluctuated slightly around zero, indicating relatively stable characteristics. The impact of the county scale factors was greater in the early stages. The development of Dongxing port not only depends on national macroeconomic policy but is also closely related to the development of Dongxing City. Dongxing developed owing to the continuous growth of the urban economic aggregate and structural transformation and upgrading that promoted the port’s rapid development. Thus, it is seen that the port drives the city’s development. The Dongxing port and city demonstrate a relationship of mutual promotion.

At the individual level, the effects of all indicators are relatively stable, mainly in the medium and long term. From 2001 to 2005, the per capita value added of the tertiary industry fluctuated the most and had positive and negative extreme values in 2002 and 2004, respectively, and then gradually approached zero. With the opening up of the large port, residents vigorously promoted the Sino-Vietnamese border trade and the city’s rapid development to pursue their own material needs and economic benefits. The port, situated 20 km north of the land border, was specified as an open point and approved by the government as a bazaar, and no more than the prescribed amount of countertrade could be carried out.

At the city level, the use of foreign capital was greatly influenced and tended to increase. The other indicators mainly demonstrated the influence of the early stages and tended to be stable in the later stages. From 2001 to 2007 and from 2017 to the present, the impact of the use of foreign capital was large, with a greater fluctuation range and multiple negative impacts. This indicates that local policies and financial support are needed in the early stage of port–city relationship development. Once the port and urban infrastructure are completed, the influence of the municipal level will gradually weaken. Therefore, the frequent economic and trade exchanges between China, Vietnam, ASEAN, and foreign capital play an increasingly important role in developing port–city relations.

At the provincial level, the influence of each index is relatively stable overall. From 2001 to 2006, the impact of the international tourism income index was relatively large and appeared negative in 2003 and 2005; however, from 2007 to now, the impact of each index gradually tended to be stable. From the provincial level, the early impact is great and mainly reflected in the economic indicators of opening up to the outside world. However, the growth rate is slow, and the stable value is low. In the early stage of port development, the policy reform on cross-border population flow and the rapid development of the international tourist numbers and industry promoted the development of ports and cities. This indicates that when a city reaches a certain stage of development, cross-border trade and the processing industry gradually develop and expand, and the impact of international tourism income on the port–city relationship decreases accordingly.

#### 3.2.4. Prediction of the Relationship between Ports and Urban Development

By representing the development indicators of the Dongxing port (number of passengers and shipments) and the urban development indicators (population and GDP), an ARIMA model was constructed to calculate the development indicator data (Table 5) and predict the value fitting diagram representing the port and city during 2020–2030.

To further analyze the relationship between the Dongxing port and the city, the cross DCI values of the predicted years were calculated according to the above Equations (1)–(3) using the predicted data of the number of passengers, shipments, population, and the GDP at the ports. The cross DCI values denoting a decade were thereby obtained (Table 6). Combined with the two groups of predicted index data representing the development of ports and cities, it consequently analyzes the development and evolution of the Dongxing port–city relationship.

Based on the findings for 2020–2030, the volume of passengers and shipments at Dongxing port will continue to grow. The cross DCI value of the shipments and population and the shipments and the GDP will gradually increase. Among them, the value of DICcp increases from 10.76 to 23.08, with an average of 17.43. The value of DICcg increases from 4.12 to 9.78, with an average value of 7.69. The freight function of the ports will gradually emerge, and port development will progressively transition to a freight-dominated development stage. Dongxing’s population will continue to grow, but its GDP will grow at a faster rate and on a larger scale. The cross DCI values of both the tourist flow and the population and the tourist flow and the GDP are stable overall while only fluctuating around the mean value. The mean values of the DICtp and DICtg are 3.62 and 1.3, respectively. With the continuous growth in the number of ports and the urban GDP of Dongxing port, the port–city relationship will transition from the port-dominated and city-dominated stages to the development stage. The port and city will co-dominate this stage, consequently reaching a new stage of balanced, high-level development of port–city relationships.

## 4. Discussion

Dongxing Port is a state-level first-class port for economic and trade exchanges between China and Vietnam. Dongxing City is a border trade city adjacent to Vietnam’s Mong Jie Special Economic Zone and the Guangxi Zhuang Autonomous Region. In 2017, China and Vietnam established the Dongxing–Mangjie Cross-border Economic Cooperation Zone to create a new growth pole for China’s border economic development. Future research, in addition to analyzing the influence factors of the national scale (China and Vietnam), should also increase the provincial scale (Guangxi Zhuang autonomous region, Quang Ninh province in Vietnam), regional scale (Shen, Mans streets), county scale (Dongxing City, Jade Ping Fang), and individual scale (residents, Jade Ping Fang) to analyze their influence [26]. 

Dongxing is the cross-border highway transportation hub of the new land-sea corridor in southwest China and plays an important role in the “Belt and Road Initiative” and the RCEP. In the future development of port–city relations in Dongxing, in addition to the influencing factors of Guangxi Zhuang Autonomous Region, the influence of economic and trade exchanges between Chongqing, Shaanxi, Sichuan, Guizhou, and Vietnam on Dongxing’s development should also be considered. Future research should also consider the impact of economic and trade exchanges with Southeast Asian countries, such as Singapore, Malaysia, Thailand, Laos, and Cambodia. The opening up of a wider market and economic hinterland will promote the development of urban transit trade and the import and export processing industry.

With the deepening of the construction of the “Belt and Road Initiative,” Dongxing city, as a window that opens up (southern) China to the outside world, has a strong momentum of development. The number of cargo and tourists passing through Dongxing port keeps increasing. In the future, to promote the port, the relationship of urban development should be achieved by developing a broader market, promoting the development of entrepot trade and import and export processing industries, conforming to transit travel sustained growth trend, integrating Dongxing city and the surrounding tourism resources, and creating enticing travel brochures [27,28], such as “go up the mountain, go into the sea, go abroad” and “exotic travel” to retain transit visitors. These measures should be followed by driving the development of the city hotels, catering, commodity trade, and other tertiary industries.

## 5. Conclusions

Based on the improved MDCI and its curve, the evolution of the Dongxing port–city relationship can be divided into three stages: the first stage (2001–2008) was characterized by low-level, balanced development; the second stage (2009–2014), port development was slightly faster than urban development, and in the third stage (2015–2019), urban development was faster than port development. The development relationship between Dongxing port and city has gradually changed from the early port-dominated stage to the city-dominated stage.

The order of multi-scale influencing factors of the Dongxing port–city relationship is country (China) > country (Vietnam) > county > individual > city > province. On a national scale, the port–city relationship is impacted by China’s economic development level and the economic and trade exchanges with Vietnam and Southeast Asian countries. The influence of the county scale is mainly reflected in the integrated development of the port and city. The individual scale reflects that individual residents promote the development of ports and cities to pursue their own economic interests. The influence of municipal and provincial scales is mainly reflected in the local financial investment needed for port construction in its early stages. After the completion of infrastructure construction, the influence of these two scales will be weakened.

The analysis of the influence and driving mechanism of different scale factors on the evolution of the port–city relationship demonstrates significant differences in the effect and direction of multi-scale factors. On the national scale, Vietnam’s industrial (agricultural) productivity, population size, and domestic economic development are the main factors affecting the port–city relationship, whereas China’s economic development and import and export of goods are the main factors. Regional economic development plays a significant positive role, whereas population size, freight volume, and inbound tourist numbers play a negative role; local scale economic development and population size play a significant positive role, whereas commodity consumption ability plays a significant negative role, and individual scale factors all play a positive role, and average annual salary, per capita disposable income, and per capita consumption expenditure has a greater impact, whereas per capita savings play a weaker role.

According to the prediction of the ARIMA model on the port–city development relationship, the growth of the port transit volume will decelerate in the Dongxing City development process during the next decade. However, the growth potential of the transit volume is great. The urban population growth rate is slow; however, the continuous growth of the GDP will drive the rapid development of the cities. The port–city relationship will gradually transition from the development stage dominated by ports and cities to the development stage co-dominated by ports and cities to realize their coordinated development. In recent years, with the increase in preferential policy support for border areas, such as the border economic cooperation zones, cross-border economic cooperation zones, and pilot projects to revitalize border areas and enrich the people, which has brought opportunities and stimulation for development to Dongxing city. However, as the COVID-19 infection continues to spread, Dongxing, a border city, is expected to be less active in the outside world. Therefore, under the influence of the promotion of preferential policies and the ongoing pandemic, the development of Dongxing’s port–city relations still faces severe challenges.

## Figures and Tables

**Figure 1 ijerph-19-09068-f001:**
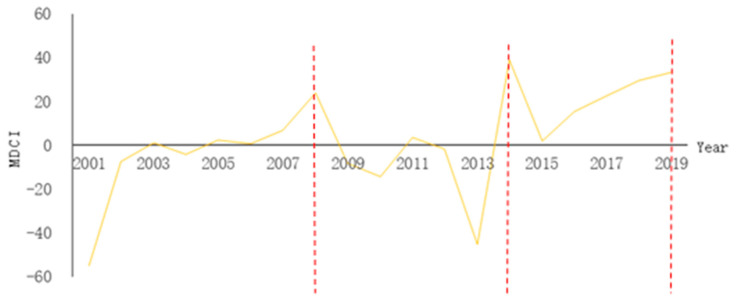
Variation curves of MDCI values in each year.

**Figure 2 ijerph-19-09068-f002:**
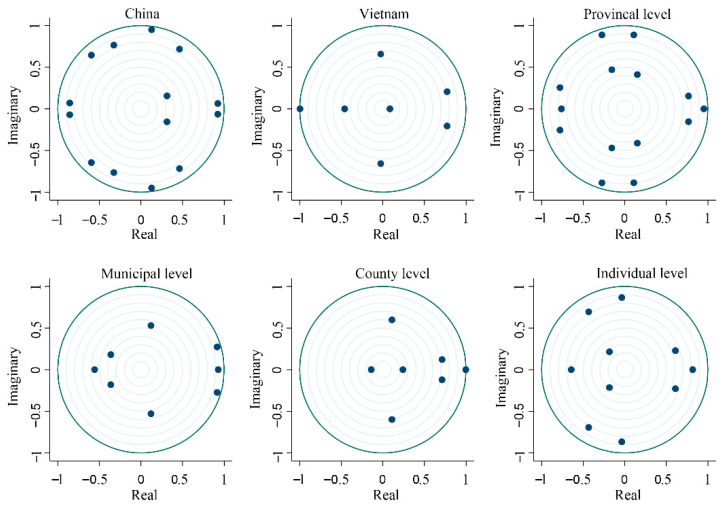
Discriminant diagram of VAR system stability at various scales.

**Figure 3 ijerph-19-09068-f003:**
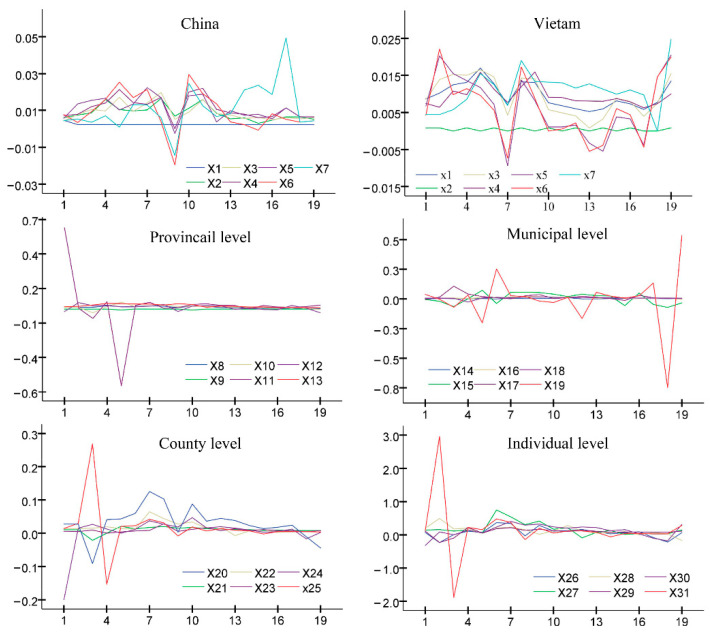
Impulse response function diagrams at different scales.

**Table 1 ijerph-19-09068-t001:** Weight of cross DCI values.

Index Items	DCIcp	DCIcg	DCItp	DCItg
Weight coefficient	14.22%	26.76%	44.51%	14.51%

**Table 2 ijerph-19-09068-t002:** Cross DCI and improved Dynamic Concentration Index at each stage.

Period	DCIcp	DCIcg	DCItp	DCItg	MDe	MDi	MDCI
2001–2008	7.69	13.65	1.27	0.28	4.44	1.05	3.08
2009–2014	7.01	18.72	0.29	2.56	3.18	5.86	4.25
2015–2019	16.59	3.04	0.12	0.80	5.57	0.58	3.11

**Table 3 ijerph-19-09068-t003:** Scale index and significance.

Scale	Indicators	Serial Number	Meaning
**Countries home**	Gross domestic product	X1	Measure the final outcome of domestic production activities
	Population	X2	Measure the total population
	Industrial growth	X3	Measure the vitality of domestic industrial production
	Household consumption expenditure	X4	Measure the vitality of domestic consumption
	Import and export amount	X5	Measure the activity of China’s foreign trade
	International tourism revenue	X6	Measure outbound tourism activity
	Agricultural income	X7	Measure the vitality of domestic agricultural production
**Province level**	Population	X8	Measure the total population of a country
	Gross regional product	X9	Measure the end result of regional production activities
	Total retail consumption	X10	Measure the consumption vitality of local residents
	International tourism revenue	X11	Measure outbound tourism activity
	Import and export amount	X12	Measure activity outside the area
	Investment in fixed assets throughout society	X13	Measure the total value of fixed assets in the region
**City level**	Population	X14	Measure the total population
	Gross regional product	X15	Measure the end result of regional production activities
	Total industrial output value of enterprises	X16	Measure the vitality of enterprise and industry
	Total retail consumption	X17	Measure the consumption vitality of local residents
	Amount of foreign capital actually used that year	X18	Measure regional openness to the outside world
	The general budget of local finance	X19	Measure spending on local public services
**County level**	Population	X20	Measure the total population
	Gross regional product	X21	Measure the final results of county production activities
	Total retail consumption	X22	Measure the consumption vigor of county area dweller
	International tourism revenue	X23	Measure outbound tourism activity
	Import and export amount	X24	Measure county area foreign trade active degree
	Investment in fixed assets throughout society	X25	Measure the total value of fixed assets in the county
**Individual level**	GDP per capita	X26	Measure the average final results of resident production activities
	Per capita fixed asset investment	X27	Measure the value of fixed assets per capita in the region
	Per capita savings	X28	Measure the income level of residents
	Retail sales per capita	X29	Measure the consumption potential of residents
	Per capita expenditure in local general budgets	X30	Measure spending on public services
	Per capita added value of tertiary industry	X31	Measure the economic status of local residents

**Table 4 ijerph-19-09068-t004:** Main component table of dimensions.

Scale	The Main Body	Kmo Test	Bartlett Sphericity Test (Approximately Chi-Squared)	Total Variance Interpretation
National scale	Vietnam	0.803	236.574	96.864
	China	0.754	291.757	98.045
Provincial scale	Guangxi	0.763	215.223	93.065
Municipal scale	Fangchenggang	0.843	266.453	93.778
County scale	Dongxing city	0.892	211.808	90.954
Individual scale	residents	0.850	311.480	92.621

**Table 5 ijerph-19-09068-t005:** Population, GDP, passenger volume, and shipment forecast from 2020 to 2030.

Year	Population	GDP	Port Overflow	Passenger Volume
2020	16.10	70.77	51.80	1400.29
2021	16.42	65.44	54.13	1461.98
2022	16.74	62.94	56.47	1523.66
2023	17.07	62.16	58.81	1585.35
2024	17.39	62.44	61.14	1647.03
2025	17.71	63.36	63.48	1708.72
2026	18.04	64.67	65.82	1770.40
2027	18.36	66.22	68.16	1832.09
2028	18.68	67.92	70.49	1893.77
2029	19.01	69.71	72.83	1955.46
2030	19.33	71.55	75.17	2017.14

**Table 6 ijerph-19-09068-t006:** Cross DCI values for each year.

	Population				GDP			
	**Time**	**De**	**Di**	**DCIcp**	**Time**	**De**	**Di**	**DCIcg**
Port overflow	2020	13.48	6.69	10.76	2020	3.16	5.57	4.12
	2021	12.96	11.41	12.34	2021	3.31	8.38	5.34
	2022	12.49	15.81	13.82	2022	3.40	10.58	6.27
	2023	12.07	19.92	15.21	2023	3.43	12.38	7.01
	2024	11.69	23.77	16.52	2024	3.44	13.89	7.62
	2025	11.34	27.39	17.76	2025	3.42	15.18	8.13
	2026	11.01	30.81	18.93	2026	3.39	16.30	8.56
	2027	10.71	34.04	20.05	2027	3.36	17.28	8.93
	2028	10.44	37.10	21.10	2028	3.33	18.13	9.25
	2029	10.18	40.02	22.12	2029	3.29	18.89	9.53
	2030	9.95	42.79	23.08	2030	3.26	19.57	9.78
	**Time**	**De**	**Di**	**DCItp**	**Time**	**De**	**Di**	**DCItg**
Passenger volume	2020	5.12	1.70	3.75	2020	1.20	1.42	1.29
	2021	5.00	1.79	3.72	2021	1.28	1.32	1.29
	2022	4.89	1.88	3.69	2022	1.33	1.26	1.30
	2023	4.79	1.96	3.66	2023	1.36	1.22	1.30
	2024	4.69	2.04	3.63	2024	1.38	1.19	1.31
	2025	4.60	2.12	3.61	2025	1.39	1.18	1.30
	2026	4.52	2.19	3.59	2026	1.39	1.16	1.30
	2027	4.44	2.27	3.57	2027	1.39	1.15	1.30
	2028	4.37	2.33	3.56	2028	1.39	1.14	1.29
	2029	4.30	2.40	3.54	2029	1.39	1.13	1.29
	2030	4.24	2.46	3.53	2030	1.39	1.13	1.28

## Data Availability

The data are available in a publicly accessible repository that does not issue DOIs. Publicly available datasets were analyzed in this study, and these data can be found here: https://navi.cnki.net/knavi/yearbooks/YINFN/detail?uniplatform=NZKPT (accessed on 3 June 2021).

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
