# Peer review of "Multi-Scale Influencing Factors and Prediction Analysis: Dongxing Port–City Relationship"

_ijerph, 2022, doi:10.3390/ijerph19159068_

Round 1
Reviewer 1 Report
The manuscript is well written. I only have a few comments.
References must be numbered in order of appearance in the text
Ln 233, 241: The software mentioned should be in Materials & Methods section
Ln 228: Make sure thal all methodology used (eg. stationarity test) is clearly mentiond in Materials & Methods section
Ln 425: authors
Author Response
Dear Editors and Reviewers:
Thanks for your letter and the reviewers' constructive comments concerning our manuscript entitled”Multi-Scale Influencing Factors and Prediction Analysis: Dongxing Port-City Relationship “(ID: ijerph-1796097). Those comments are all valuable and very helpful for revising and improving our manuscript, and significantly guide our research. We have studied these comments carefully and made modifications which we hope meet with your approval. Revised portions are marked in the manuscript. The main corrections in the manuscript and the responses to the reviewers' comments are listed below.
Response 1:
We are very grateful to reviewer for the effort he/she put into reviewing our paper and for his/her positive feedback. The summary of the work written by this reviewer is specific and helpful. Our resolutions of the questions and suggestions raised by reviewer are listed as follows:
- The references have been revised as requested in accordance with the reviewer's suggestions. References are numbered in the order in which they appear in the text. (See References for details)
- As suggested by the reviewers, we have revised the software method and placed it in the 2.2. Research Methods section.
- As suggested by the reviewers, we have ensured that all methods used, such as modified dynamic concentration indices, impulse response functions based on VAR models, principal component analysis (PCA), smoothing test models and autoregressive differential moving average models (ARIMA), are explicitly mentioned in the Materials & Methods section. (See 2.2. Specific study methods for details)
- We've changed author singular to plural. “Conflicts of Interest: The authors declares no conflict of interest.”

Reviewer 2 Report
In the first instance I ascertained that the article met the requirements of the Journal and Academia in general. My findings were that this article has been written for a Special Edition of the MDPI Journal “International Journal of Environmental Research and Public Health ISSN 1660-4601” and meets the requirements stipulated by the Editors of the Special Edition. It has been researched and written by a team of researchers and so also meets the requirements of collaboration. This collaboration is international and that is a significant bonus. The references in the article indicate that primary research has been conducted and presented and discussed in a peer environment including workshops. Reference is made to secondary literature and so this article builds upon and critiques other research. Given the above I turned to review the content to ascertain if it met the requirements of being innovative, in contributing to knowledge and in analysis that would place it firmly as worthy of publication. My findings were that the article meets these requirements. A short summary of my findings are as follows.
This paper is significant as it comes at a time of the deepening of China’s national strategic concept of the “Belt and Road Initiative” to promote endorsement of the 36 Regional Comprehensive Economic Partnership agreements (RCEP). Furthermore, domestic and international strategies concerning frontier ports and cities have also shifted. China is a major actor in global trade and also it’s borders with 14 countries on land and faces eight countries by sea. As of December 31, 2020, the State Council had approved the opening of a total of 313 ports to the outside world, including 82 highway ports, 129 water ports, 22 railway ports, and 80 air ports. Land border areas have 140 ports open to the outside world, of which 70 are highway ports, 34 are water ports, 13 are railway ports, and 26 are air ports.
This study takes one example of these. It calculates the dynamic concentration index, explores the evolution of the relationship between the Dongxing port and city, and predicts its future. It is a scholarly paper building upon the knowledge in existing literature. For example much domestic and foreign literature has studied the port-city relationship. The contribution of this paper to this literature is to explore the evolutionary mechanism and suggest new ideas to promote the sustained and steady development of the China-Vietnam border trade.
The results indicate that the relationship between the port and city has three development stages, namely the low-level balanced development stage (2001–2008), the port development stage (2009–2014), and the urban development stage (2015–2019). Based on the country (China and Vietnam), province (Guangxi Zhuang Autonomous Region), district (Fangchenggang City), county (Dongxing City), and individual (resident) levels, a multi-scale index system of influencing factors was developed.
The impulse response function model analyzed the influential factors in the relationship between port and city development. The influence is as follows: Country (China) > Country (Vietnam) > County > In-dividual > City > Province. Finally, the relationship between port and city development was predicted using an auto-regression differential moving average model. It is expected that Dongxing City will gradually transition from a port- and city-dominated stage to a new stage of coordinated development. Thus, by improving the proportion of the secondary and tertiary industries, managing the population density, introducing foreign capital, enhancing the innovation level, and improving the traffic facilities, high-quality development in Dongxing port-city can be achieved.
This paper makes a contribution to the field in the presentation of a theoretical model, a review of literature, and the discovery of new information through original research. The existing research mostly analyses the evolution of the port-city relationship on a single scale, whereas the present study systematically analyses the mutual-feed mechanism and dynamic simulation of the port-city relationship from a multi-scale perspective. Some policy suggestions are presented for the coordinated development of the Dongxing port and city. The methodology for example includes to systematically collect the economic and social development data of Dongxing port and Dongxing City. The research method associated with this was to refer to the Modified Dynamic Concentration Index (MDCI) to represent the relationship between port and urban development.
Substantial primary date in tables and figures sustained the thesis of article proving the case beyond doubt. The bottom line of the findings was that the port-city relationship will gradually transition from the development stage dominated by ports and cities to the development stage co-dominated by ports and cities to realize their coordinated development. However, some cautions were also voiced that the growth potential of the transit volume is great. The urban population growth rate is slow; however, the continuous growth of the GDP will drive the rapid development of the cities. According to the prediction of the ARIMA model on the port-city development relationship, the growth of the port transit volume will decelerate in the Dongxing City development process during the next decade.
Strengths
The key to success in writing this paper is critical thinking. The task of every author of a research article is to convince readers of the correctness of his or her viewpoint, even if it is skewed and this has been achieved. Thus, the only ways to distinguish solid arguments from weak ones are to be a good researcher, have the right tools to pick out facts from fiction, and possess solid critical thinking skills. The strength of this paper is noted on this point. It is worth noting that the main purpose of this critique is to bring up points that determine whether the reviewed article is either correct or incorrect. First of all, in my process of ascertaining the critique of the article, I have considered if the authors are experts is the expert in their field and concluded they are. They are knowledgeable about the topic and are able to convey a main message that is clear. There are more than just general phrases with any specific details.
The sources used by the author are multiple without any logical blindspots and arrives at a clear outcome in his or her work. There are logical fallacies. That include Ad hominem, Slippery Slope, and Correlation vs. Causation without the presence of biased opinions. That is to say the argument is based on evidence rather than the outcomes. There is no evidence that the author overstates the importance of some things due to his or her own beliefs. The authors cited trustworthy sources of information. Just like any other written assignment, a critique paper should be formatted and structured properly and this was done. A standard article consists of four parts: an introduction, summary, critique, and conclusion and this was followed as noted beelow
Introduction
• The core idea of the author.
• A clear thesis that reflects the direction.
• Summary
• The main idea of the article.
• The main arguments presented in the article.
• The conclusion of the article.
• Critique
• Express an educated opinion regarding the relevancy, clarity, and accuracy of the sources
• Conclusion
• Summary of the key points of the article.
• Finalization of conclusion with comments on the relevancy of the research.
Language plays a vital role in every article, regardless of the field and topic and here the use of such language has been a clear sign of supported fallacies. In considering a checklist
• How is the design of the study? There were no errors.
• How does the piece explain the research methods? Yes
• Was there a control group used for this research? Yes
• Were there any sample size issues? No
• Were there any statistical errors? No
• Is there a way to recreate the experiment in a laboratory setting? yes
• Does the research (or experiment) offer any real impact and/or value in its field of science? Yes
A weakness
The article is geared towards a specialist audience and appeals to a specific group of people using language that is only understandable to that audience
Author Response
Dear Editors and Reviewers:
Thanks for your letter and the reviewers' constructive comments concerning our manuscript entitled”Multi-Scale Influencing Factors and Prediction Analysis: Dongxing Port-City Relationship “(ID: ijerph-1796097). Those comments are all valuable and very helpful for revising and improving our manuscript, and significantly guide our research. We have studied these comments carefully and made modifications which we hope meet with your approval. Revised portions are marked in the manuscript. The main corrections in the manuscript and the responses to the reviewers' comments are listed below.
Response 1:
It is an honour to receive the approval of the reviewers and your detailed comments and sincere opinions on our manuscripts are greatly appreciated. Each review of my thesis has brought great help to the revision of my thesis and subsequent scientific research work.
As to what you have mentioned about “the article is geared towards a specialist audience and appeals to a specific group of people using language that is only understandable to that audience”. I have revisited the language used in the article, sought to understand the language, and revised it. See the revised section for more details.

Reviewer 3 Report
The paper addresses a predictive analysis of a port-city relationship in Dongxing City. The paper is well structured and the methodology correctly drafted and explained. If the authors are interested, I would only suggest to provide additional limitations in the conclusions. As a merely quantitative approach, this could leave out some impactful aspects such as social, cultural, historical ones. How is the port-city developing relationship affected the citizenship or local workers? Where there some side effect (e.g. population expulsion or gentrification mechanisms)?
I understand this is not the main focus of the paper, but to fit some reflections in the conclusions could add some perspective on usability of this study and replicability in other contexts. I also suggest two publications of this matter: https://doi.org/10.6092/issn.2612-0496/v4-n1-2021; https://doi.org/10.6092/issn.2612-0496/v4-n2-2021
Author Response
Dear Editors and Reviewers:
Thanks for your letter and the reviewers' constructive comments concerning our manuscript entitled”Multi-Scale Influencing Factors and Prediction Analysis: Dongxing Port-City Relationship “(ID: ijerph-1796097). Those comments are all valuable and very helpful for revising and improving our manuscript, and significantly guide our research. We have studied these comments carefully and made modifications which we hope meet with your approval. Revised portions are marked in the manuscript. The main corrections in the manuscript and the responses to the reviewers' comments are listed below.
Response 1:
We are very grateful to the reviewers for taking time out of their busy schedules to review our manuscript and give valuable opinions. I agree your view that as a merely quantitative approach, this could leave out some influential aspects such as social, cultural, historical ones. Based on the reviewers’ suggestions and the two references provided above, we reviewed this part again and made the following modifications:
As a border area, the development of port cities takes into account the role of policies of border areas, which will bring development opportunities to border areas. Meanwhile, the impact of COVID-19 on port cities cannot be ignored.
In recent years, with the increase of preferential policy support for border areas, such as the border economic cooperation zones, cross-border economic cooperation zones, and pilot projects to revitalize border areas and enrich the people, which has brought op-portunities and stimulation for development to Dongxing city.. However, as the COVID-19 infection continues to spread, Dongxing, a border city, is expected to be less active in the outside world. Therefore, under the influence of the promotion of prefer-ential policies and the ongoing pandemic, the development of Dongxing’s port-city re-lations still face severe challenges.